# Bipolar electrochemical tweezers using pristine carbon fibers with intrinsically asymmetric features

Bhavana Gupta [1,2] ✉, Vishal Shrivastav [1,3], Shashank Sundriyal[3,4], Ambrose Ashwin Melvin[5], Marcin Holdynski [1], Alexander Kuhn [6] & Wojciech Nogala [1] ✉

Structures that can be stimulated to change shape may be utilized for a variety of applications, but they frequently need to be processed and modified. We propose here a simple, straightforward strategy of actuation based on bipolar electrochemistry driving asymmetric reactions at the surface grooves of pristine carbon fibers. In the first set of proof-of-principle experiments, a free-standing carbon fiber is polarized in a closed bipolar cell to trigger asymmetric benzoquinone/hydroquinone redox reactions in the two distinct compartments. Beyond a particular threshold potential, ion transfer occurs, and the part of the fiber involved in the anodic reaction exhibits reversible directional motion. Elemental surface characterization of the polarized carbon fiber indicates that the deflection is due to the intercalation/deintercalation of ions accompanying the oxidation/reduction of the fiber. The simultaneous local surface ionic adsorption/desorption is responsible for the fiber deflection. The length of the fiber part exposed to the electrochemical reduction reaction in the opposite compartment of the closed bipolar cell, as well as the groove orientation, determines the motion's intensity and direction, respectively. Effective bending is achieved by optimization of fiber alignment and stimuli parameters. Actuation of two parallel fibers, oriented in opposite directions, leads to microtweezer-type behavior. We anticipate that these results will enrich the tool case for research in the field of soft robotics and micromechanics.

Low-density structures like conducting polymers and carbon materials that can change their shape when exposed to an electric field are valuable for many applications ranging from soft robotics to biomedical devices and micro-electromechanical systems[1–3]. These materials can produce high actuation strain at high frequency under low operating voltage, much like a genuine muscle, because of pseudo-capacitance phenomena[2,3]. In the case of carbon materials, a unique approach is employed to decrease the operating voltage, triggering an electrochemical response that is useful for actuation. Baughman et al. demonstrated actuation in an ensemble of single-walled carbon

[1]Institute of Physical Chemistry, Polish Academy of Sciences, Kasprzaka 44/52, 01-224 Warsaw, Poland. [2]Department of Chemistry, Cluster of Applied Sciences, School of Advanced Engineering, UPES, Dehradun, Uttarakhand, India. [3]Regional Center of Advanced Technologies and Materials, Czech Advanced Technology and Research Institute (CATRIN), Palacký University, Olomouc, Czech Republic. [4]Electrical Cluster, School of Advanced Engineering, UPES, Dehradun, Uttarakhand, India. [5]Department of Chemical & Biomolecular Engineering, Sogang University, Seoul, Republic of Korea. [6]Univ. Bordeaux, CNRS, Bordeaux INP, ISM UMR 5255, 33607 Pessac, France. ✉e-mail: bhavana.gupta@ddn.upes.ac.in; wnogala@ichf.edu.pl

nanotubes due to quantum chemical-based expansion caused by electrical double layer charging in a 1 V operational range[4]. Kong et al. obtained carbon materials and an ionic polymer metal composite (IPMC) actuating optimally at 3 V[2]. To trigger actuation in response to an applied potential of 1 V, Liang et al. synthesized graphene/$Fe_3O_4$ hybrid paper[5]. $Fe_3O_4$ nanoparticles were employed to partially inhibit graphene sheets from restacking and facilitate the electrolyte ions to penetrate the resulting magnetic graphene paper to boost electrical double layer capacitance and actuator performance. All these techniques include manufacturing of bi- or tri-layers and structural modification. In contrast to real muscles, electrochemical oxidation and reduction of carbon and conducting polymer materials typically require a direct physical connection to an electric power source[6–9], which might limit the possible range of applications. Therefore, wireless electrochemical actuation of such materials constitutes an interesting alternative. Bipolar electrochemistry has recently been used for wireless actuation as a simple and very efficient method to initiate large amplitude actuation in conducting polymers[10–13]. However, the proposed method cannot be used with carbon fibers due to the higher

voltage (compared to conducting polymers) that is needed for the electrooxidation of the carbon surface, which would allow an uptake or release of ions and lead to actuation.

In the case of bipolar electrochemistry, certain experimental parameters can be optimized to lower the voltage that needs to be applied for generating a sufficient polarization of the bipolar object, such as using nanopipettes[14], shortening the distance between the feeder electrodes[15], lowering the electrolyte concentration, and employing closed bipolar electrochemical set-ups[16]. In the latter case, the oxidation and reduction compartments are physically separated. This allows for facilitating the redox processes at the conducting objects that serve as bipolar electrodes, e.g., for inducing color changes or sensing applications[17]. In addition to the generation of asymmetric features on conducting objects at low voltage, closed bipolar electrochemistry might also be very advantageous to induce actuation, but this has not been investigated yet.

Asymmetric bi-layer and tri-layer structures are a crucial requirement for electrochemical actuation and are typically generated by surface modification or other special fabrication techniques, which might be time-consuming and challenging[18–20]. However, in the case of conducting polymer films, asymmetrical structures that show actuation in an open bipolar electrochemical cell can be easily generated via a bottom-up manufacturing approach[10]. Similar asymmetric features might also allow the actuation of carbon fibers, but due to their axial symmetry, this is more difficult to achieve. Foroughi et al. used a three-electrode electrochemical setup to design an intrinsically inhomogeneous twist-spun carbon nanotube yarn that may act as a torsional artificial muscle, offering reversible multiple revolutions[21]. However, analog actuation experiments with single carbon fibers have not been reported so far probably because of missing asymmetric features, which is a key ingredient for actuation. Surprisingly, the mechanical performances of carbon fibers are significantly influenced by their surface features. The development of rough topographies or grooves on the surface of wet-spun carbon fibers severely restricts their strength. However, such surface features can also be an ingredient for enhancing electrochemical reactions due to a larger surface area and can also be a source of asymmetry if the grooves are not uniform[22,23]. Due to the inherent structural asymmetry, these irregular surface grooves offer a potential for electrochemical actuation.

In this contribution, we suggest using surface features of carbon fibers as a crucial ingredient of asymmetry to induce their directional motion/actuation with the help of bipolar electrochemistry at reduced operating potentials. Finally, this type of actuation allows the carbon fibers to operate as bipolar electrochemical tweezers.

## Results and discussion
To achieve efficient directed movement of a carbon fiber, it is crucial to break the symmetry of the system, primarily in terms of surface morphology. Two types of carbon fibers were employed to investigate their surface morphology. SEM was used to characterize the surface to confirm asymmetric features. In one type of carbon fiber, the surface of the fiber does not appear to be smooth but shows irregularities like grooves (Fig. 1a–c). The cross-section image of the fiber fused in a glass capillary, followed by polishing (Fig. 1c), clearly illustrates an asymmetric distribution of these grooves. One can observe that the lower half of the fiber has a noticeably different groove geometry than the upper half. Consequently, the lateral microscopic surface area on one side of the fiber differs from that on the other side. Moreover, the cross section of a fiber exposed not by polishing of a composite with a glass capillary, but by mechanical cutting (Fig. 1g), shows a clear asymmetric pore distribution, which is substantially different from the cross section of a smooth fiber. The latter one is compact and symmetrical (Fig. 1h). The ions from the supporting electrolyte can be taken up asymmetrically by the carbon fiber polarized at positive potentials. For the second type, grooves are visible in SEM images but are regularly

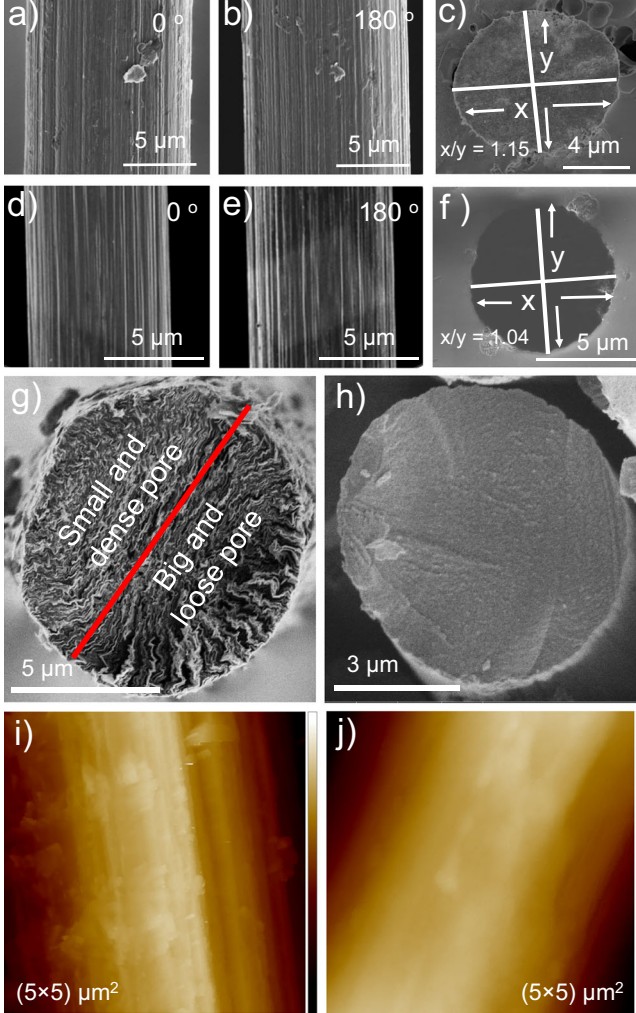

**Fig. 1 | Surface characterization of rough and smooth carbon fibers.** Scanning electron microscopy images of a carbon fiber surface for 0° and 180° axial rotation of a rough carbon fiber (**a**, **b**) and a smooth carbon fiber (**d**, **e**); cross-section view of a carbon fiber fused in glass capillary and exposed by polishing: (**c**) a rough carbon fiber, and (**f**) a smooth carbon fiber. Figures (**g**, **h**) show cross-sections of rough and smooth carbon fibers, respectively. **i** AFM image of a rough carbon fiber (z-scale: 836 nm), and (**j**) AFM image of a smooth carbon fiber (z-scale: 2.4 μm). Cross sections of AFM images are presented in the Supplementary Fig. 1.

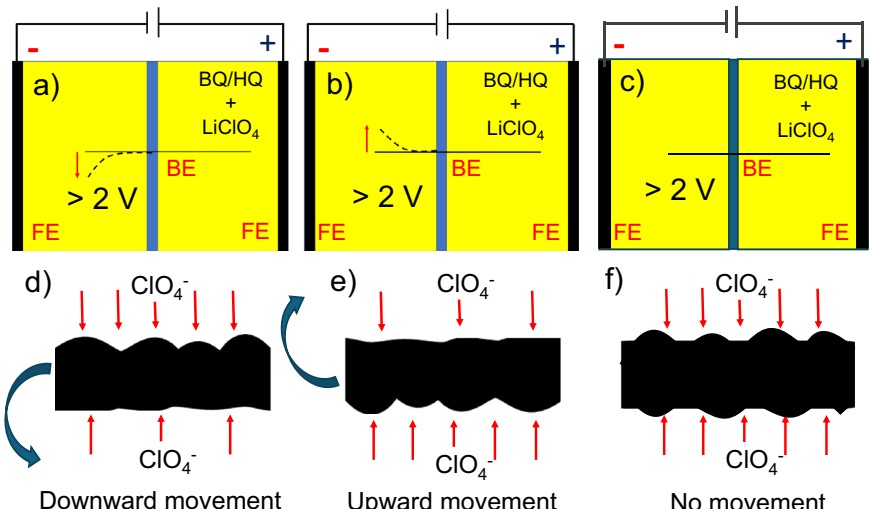

**Fig. 2 | Carbon fiber actuation mechanism.** Schemes of the closed bipolar electrochemical cell used for carbon fiber actuation for (**a**–**b**) an asymmetrically rough fiber with upward and downward motion (**c**) smooth fiber without motion (**d**–**e**) asymmetric ionic movement inducing downward and upward movement and (**f**) symmetric ionic movement at the carbon fiber. BQ benzoquinone, HQ hydroquinone, FE feeder electrode, BE bipolar electrode.

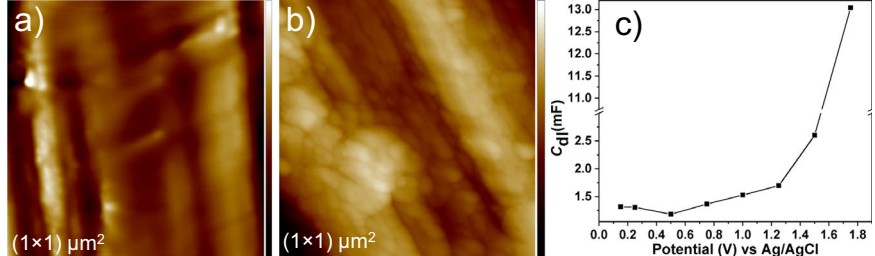

**Fig. 3 | Surface modification of carbon fibers by electrochemical oxidation.** AFM image of an asymmetric carbon fiber (**a**) before (z-scale: 28.3 nm) and (**b**) after bipolar electrochemical oxidation from 0–3 V in 10 mM BQ/HQ and 0.25 M LiClO₄ (z-scale: 138.6 nm). **c** Electrical double layer capacitances ($C_{dl}$) derived from impedance spectra recorded with a bundle of fibers at various DC potentials. Solid lines are a guide for the eye. Source data are provided as a Source Data file.

arranged all along the surface (Fig. 1d–f). The symmetry in terms of shape is also not the same for the two types of fibers. In the first case, the x/y ratio is higher compared to the second type, which constitutes an additional ingredient of asymmetry. Additionally, AFM was carried out to investigate the microscopic features of the grooves. The rough fiber possesses clearly visible nonuniform surface features (Fig. 1g) in comparison to the rather uniform surface of a smooth fiber (Fig. 1h).

To investigate the carbon fiber actuation, both types of fibers were positioned in a closed bipolar electrochemical setup, containing LiClO₄ as a supporting electrolyte and benzoquinone (BQ) and hydroquinone (HQ) as a redox couple, as depicted in Fig. 2. To detect the motion of the carbon fiber, it was positioned in the focus of a video microscope. The length of the fiber participating in the electrooxidation is 4.5 mm, whereas the other side is 9 mm long. Once a voltage of more than 2 V is applied to the feeder electrodes, a progressive displacement of the fiber, resembling the movement of an arm, can be seen in the case of the carbon fiber having non-uniform grooves on the surface (Fig. 2a, b), while the carbon fiber with uniform groves does not move (Fig. 2c). An influence of convection on the movement of the fibers can be excluded because actuation does not occur in the case of a non-conductive object with a comparable structure, such as a human hair (Supplementary Video 1). ClO₄⁻, the anion originating from the LiClO₄ salt present in solution, gets adsorbed asymmetrically on the surface of the carbon fiber polarized at positive potentials during the formation of the electrical double layer. At the fiber face presenting larger grooves, more ions interact with the carbon fiber, which results in a directional motion analog to the wireless actuation observed for conducting polymers[10]. With the increase in potential, it is assumed

that the porosity increases, which is responsible for the larger value of double-layer capacitance, as shown in Fig. 3c. This feature is further confirmed by electrochemical impedance spectroscopy measurements at different potentials. It was found that an increase in double-layer capacitance with an increase in electric potential is due to an increase in porosity. This feature is further confirmed by AFM measurements before and after the electrochemical actuation of carbon fiber, as shown in Fig. 3a, b. The surface roughness of the carbon fiber increases after electrochemical oxidation. Nanoscale roughness factors before and after oxidation are 1.01 and 1.06 nm, respectively. The direction of displacement varies from fiber to fiber as a function of their axial rotation. In-situ voltammetric studies with a reference electrode (Ag/AgCl) were conducted to better understand the actuation mechanism during the electrochemical process leading to the deflection of the fiber (Fig. 4). The total deflection of the carbon fiber is 0.55 mm (Supplementary Fig. 2) from 2 V to 3 V. Maximum deflection occurs at 3 V. Above 3 V, oxygen bubble generation takes place, which creates ambiguous deflection. In the voltammograms, two steps of rising current can be clearly identified. Both compartments contain the same electrolyte, and both redox forms (hydroquinone and benzoquinone) are present at identical concentrations. When oxidation of hydroquinone to benzoquinone is driven on the shorter fiber in the left compartment (anodic current), then the opposite process (cathodic, reduction of benzoquinone to hydroquinone) occurs at the longer fiber in the right compartment. The magnitude of the current is the same in both compartments, however, current densities differ due to unequal lengths (areas) of the two fiber fractions. When the oxidation of hydroquinone to benzoquinone reaches a diffusion-limited rate in

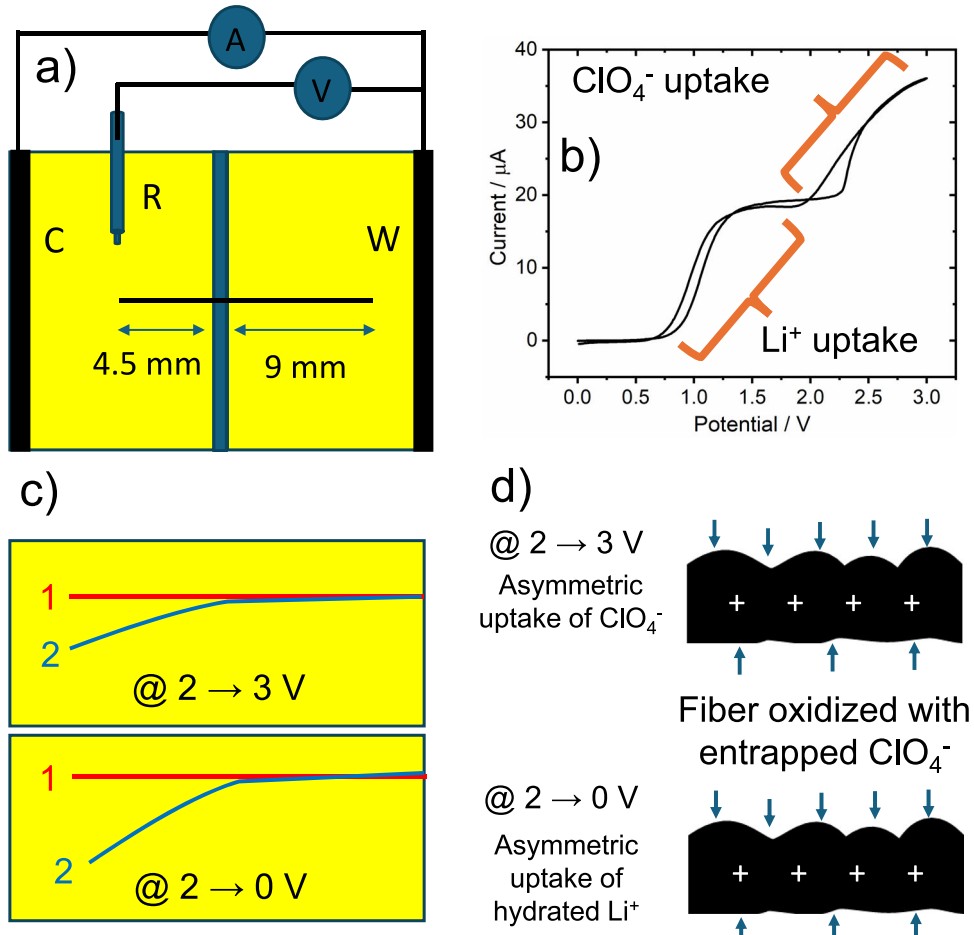

**Fig. 4 | Voltammetry and deflection of a carbon fiber during electrochemical stimulation. a** A scheme of the three-electrode set-up (electrodes: C – counter, R – reference, W – working) in a closed bipolar electrochemical cell used for carbon fiber voltammetry studied during the actuation. **b** Voltammograms of a carbon fiber during its upward/downward deflection with ionic uptake in a specific potential region, (**c**) deflection at the different voltage transitions (2–3 V and 2–0 V). Positions 1 (red) and 2 (blue) are before and after actuation, respectively. **d** Corresponding scheme of the ionic transfer process in and out of the electrical double layer. Source data are provided as a Source Data file.

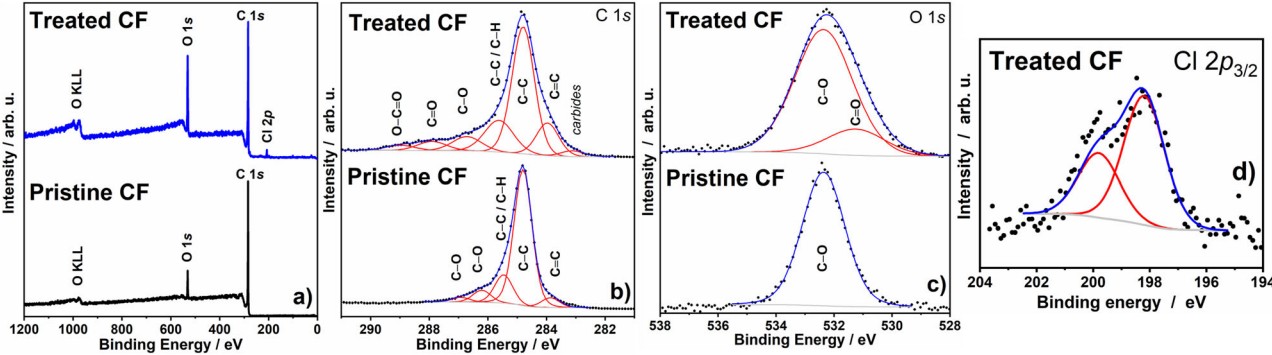

**Fig. 5 | XPS analysis of pristine and electrochemically treated carbon fibers. a** Survey (**b**) C1s (**c**) O1s XPS spectra of a pristine and treated carbon fiber (CF), followed by deconvolution, and (**d**) Cl 2p spectra of a treated carbon fiber. The fitting results are summarized in Supplementary Table 1. Source data are provided as a Source Data file.

the left compartment, the reduction rate of benzoquinone in the right compartment is still below the diffusion-controlled limit. Thus, another faradaic process must occur in the left compartment to increase the current and the next available anodic process is the oxidation of water. Therefore, the next wave, starting above 2 V, corresponds to the oxidation of water on the carbon fiber in the left compartment, whereas the reduction of benzoquinone still occurs on the longer fiber segment in the right compartment until it reaches the diffusion limit represented by a second plateau in the recorded voltammograms (Fig. 4b). In general, contrary to water oxidation, electrooxidation of hydroquinone to benzoquinone and electroreduction of benzoquinone to hydroquinone occurs on carbon materials at low overpotentials[24,25]. However, water oxidation occurs at a potential high enough to drive also the oxidation of the carbon fiber surface. Oxidation of carbon in the left compartment accompanied with electroreduction of benzoquinone to hydroquinone in the right compartment

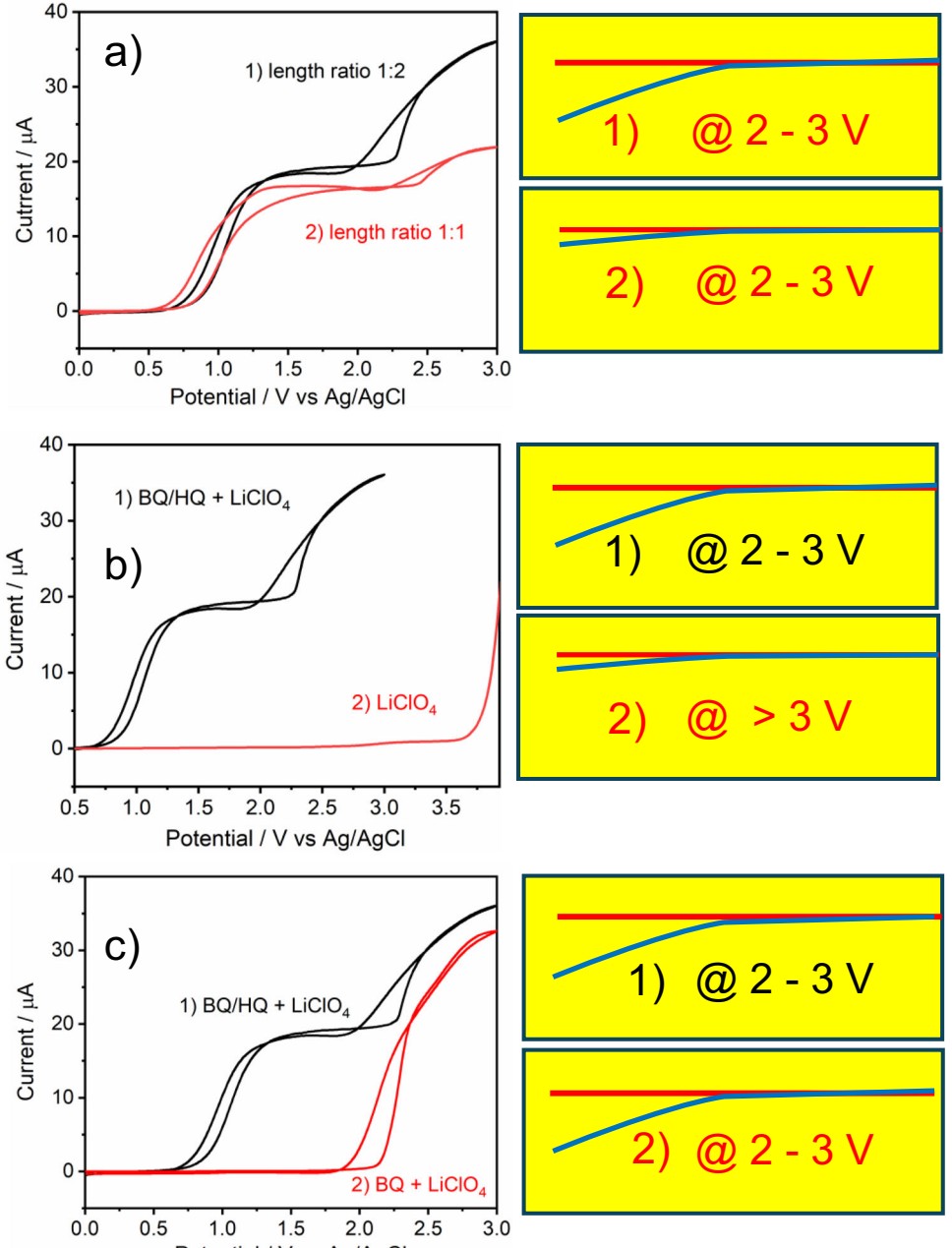

**Fig. 6 | Effect of fiber length ratio and electrolyte composition on voltammetry and actuation.** Cyclic voltammograms recorded under three different conditions and compared with standard voltammograms obtained with a 1:2 length ratio of the carbon fiber segments in the anodic and cathodic compartments containing $LiClO_4$ with benzoquinone (BQ) and hydroquinone (HQ) (1). The three different conditions are: (**a**) 1:1 length ratio of the carbon fiber in cathodic and anodic compartments, i.e., 4.5 mm (2); (**b**) only $LiClO_4$ supporting electrolyte in both compartments (2); (**c**) without hydroquinone (only benzoquinone) in both compartments (2). Graphics 1-2 are the actuations corresponding to the voltammogram. Positions of the fiber before and after actuation are marked red and blue, respectively. Potentials are marked in the figures. Source data are provided as a Source Data file.

induces the actuation between 2 and 3 V (Fig. 4b and Supplementary Video 2, 3). Besides the oxidation of the surface of the carbon fiber in the left compartment causing a change of the surface structure, the surface of the fiber gets a high positive charge at such a high potential. This positive charge must be compensated by the insertion of available anions into the electrical double layer (EDL). Due to the irregular grooves, $ClO_4^-$ ions enter the EDL at the carbon fiber asymmetrically and neutralize the positive charge from 2 V to 3 V, causing directional deflection (Supplementary Videos 2, 3). Another intriguing feature can be seen during the reverse potential sweep. In the case of reversible charging-discharging of the EDL, only $ClO_4^-$ ions would desorb from the fiber, causing recovery of its initial position during the voltage scan

from 3 V to 2 V. However, re-actuation is observed when the voltage is further decreased from 2 to 0 V. The position of the fiber is not fully recovered, probably because of thermodynamically irreversible oxidation of the carbon fiber, causing hysteresis. $Li^+$ ions can enter the EDL with entrapped $ClO_4^-$ ions during the reverse sweep (from 2 V to 0 V), resulting in an increase of deflection higher than the one originating from the initial insertion of $ClO_4^-$. A similar behavior was observed in the case of multiwall carbon nanotube fibers, where anions incorporated into the fiber stayed during discharging, triggering cation addition to the EDL, and the fiber showed mixed actuation[26]. The concentration of $LiClO_4$ does not affect actuation and current (Supplementary Fig. 3). The magnitude of actuation depends on the applied

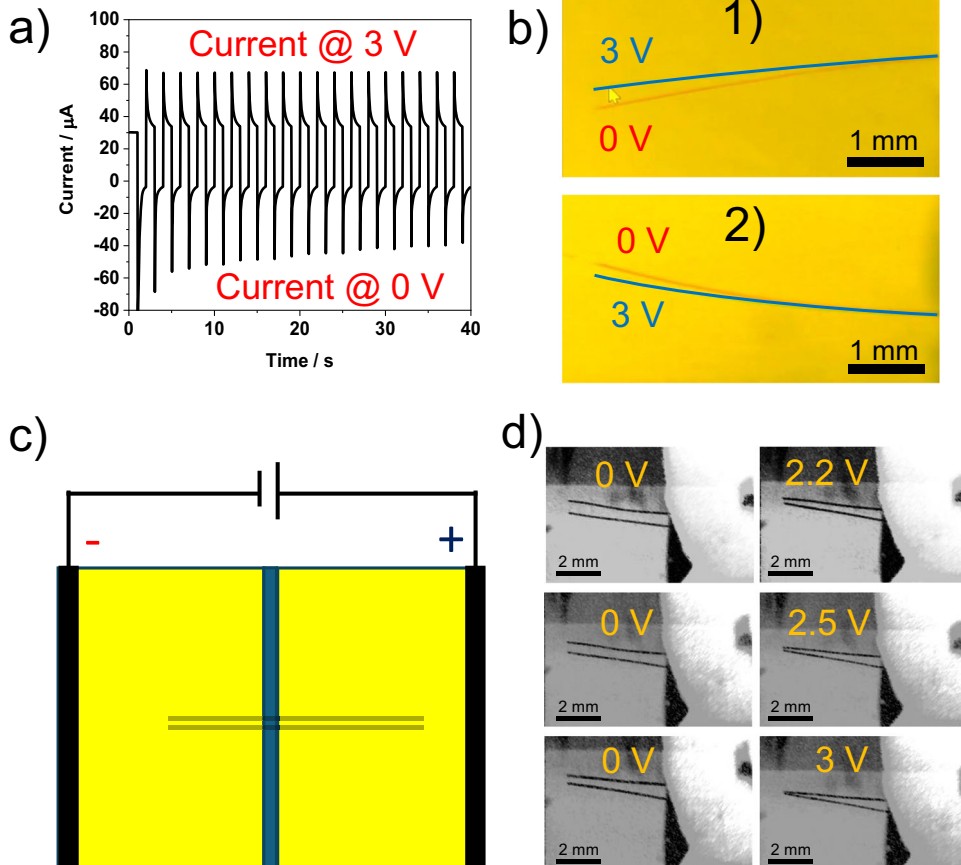

**Fig. 7 | Electrochemical actuation dynamics of carbon fibers and micro-tweezers. a** Current vs time curve for the pulse 0-3 V for 10 cycles. **b** Illustration of the fiber position at 0–3 V for a fiber 1) moving downwards in the first voltammetric cycle and 2) moving upwards in the first voltammetric cycle. Positions of the fiber polarized at 0 V and 3 V are marked red and blue, respectively. **c** Closed bipolar setup with two parallel carbon fibers. **d** Illustration of micro-tweezers at different potentials. Source data are provided as a Source Data file.

potential and the length of the fiber. Strong actuation also occurs at voltages lower than 0 V (Supplementary Video 3), which further confirms the role of Li$^+$ ions uptake.

If the fiber goes upward for voltages 2–3 V during the first forward sweep, then it moves in the opposite direction (downward) during the following forward sweep. A voltammetric first forward cycle actuates the fiber by ClO$_4^-$ ion insertion, and the subsequent cycles are both regulated by hydrated Li$^+$ ions and ClO$_4^-$ ions. To better understand the proposed ion insertion process, it is necessary to study the oxidation state of the carbon fiber. Thus, XPS characterization was performed as shown in Fig. 5.

In the same electrolyte as the one used for the actuation, two samples were examined with XPS before and after 10 cycles of voltammetry in the 0 to 3 V potential range. After the electrochemical treatment, the primary carbon peaks are shifted to higher binding energies, revealing a marked variation in oxidation state[27–29]. Several additional peaks also appeared at higher binding energies, further supporting an oxidized state of carbon. Two highly energetic states with energies between 198.2 and 199.8 eV can be seen in Cl XPS spectra, and they are only present for samples after the electrochemical test. A pure (non-treated) sample does not exhibit any Cl element signal. This finding confirms that ClO$_4^-$ ions participate in the electrochemical reaction taking place during actuation. There is an additional peak at 199.8 eV in the Cl XPS spectra (Fig. 5d), resulting from C-Cl bond formation. The results of the XPS analysis unequivocally demonstrate that carbon oxidation occurs during the activation of the Faradaic process at the carbon fiber.

Several experiments were conducted to verify the impact of the reduction current of benzoquinone to hydroquinone transformation in the right compartment. To start, the length of the carbon fiber in the right compartment was cut to 4.5 mm, resulting in a length ratio of the two compartments of 1:1. In this scenario, both the degree of deflection and the current (second increase) were reduced dramatically compared to the sample with a length of 9 mm (Fig. 6a). This demonstrates unequivocally how current affects actuation. Analyte concentration also affects the current that triggers actuation. If benzoquinone and hydroquinone are removed from the solution, the first and second current waves vanish from the voltammogram. Simultaneously, the voltage for actuation increases to above 3 V and current also starts to flow at 3 V when no redox mediators are present in the electrolyte (Fig. 6b).

This serves as additional evidence that water oxidation coincides with the actuation. In addition, hydroquinone was removed from the left-side chamber to avoid its oxidation to benzoquinone, causing the initial surge in current. The second rise in current (here the first) corresponds to water oxidation in the left compartment, but is limited by the diffusion-limited reduction of benzoquinone to hydroquinone in the right compartment, which persists at the same voltage (ca. 1.9 V) and also results in actuation with the same intensity (Fig. 6c). This demonstrates unequivocally that oxidation of water coincides with actuation and its magnitude is limited by the cathodic process in the right chamber.

Since the fiber movement is controllable, it can be used to demonstrate several interesting features. It is evident from the

voltammogram that the fibers tend to migrate in one direction at one voltage and partially return to their initial location at a different voltage. The direction of the reversible actuation is determined by the direction of the actuation during the first voltammetric cycle. For instance, if a fiber moves up in the first voltammetric cycle, it moves down at 3 V, and then back up at 0 V. The opposite is observed when the fiber moves downwards in the first potential cycle. When staying in the same potential window, this up-and-down movement repeats for multiple cycles without variations in the amplitude of actuation (Fig. 7a, Supplementary Video 4). The amplitude of actuation is smaller for shorter potential windows. Similarly, a 2-s pulse (apply 3 V stay there for 2 s and then jump back to 0 V) causes more actuation than a 1-s pulse. Finally, two fibers were fixed parallel to each other inside a closed bipolar cell, as shown in Fig. 7c. The degree of deflection is controlled by applying a variety of voltage windows. The two fibers move towards one another when a 0–3 V potential window is applied, eventually touching like tweezers[30]. Multiple cycles of the back-and-forth movement feature showed the successful micro tweezer development (Supplementary Video 5).

In conclusion, we successfully used the closed bipolar cell to wirelessly actuate a freestanding carbon fiber electrochemically. An uneven electrical double layer is made possible by their naturally asymmetric groove configuration, which is one of the fundamental factors in producing the necessary initial asymmetry. An electric field in the electrolyte solution causes the second asymmetry by allowing simultaneous oxidation and reduction reactions in the two compartments of the bipolar cell. This leads to an asymmetric driving force around carbon fiber. The mechanism of actuation, as well as the details of the involved local electrochemical activity, have been studied to demonstrate how two fibers can move towards and away from one another like a pair of tweezers. This is the first set of proof-of-concept tests showing how bipolar electrochemistry may initiate motion in pristine carbon fibers under the control of an electrochemical reaction when asymmetric surface features are present. As a result, these findings may open up intriguing possibilities for actuators based on prefabricated asymmetric carbon fibers.

## Methods

Bipolar electrochemical actuation was carried out in a solution containing $LiClO_4$ (0.25 M), benzoquinone (10 mM) and hydroquinone (10 mM) obtained from Sigma-Aldrich. The rough carbon fibers of 10 μm diameter and smooth ones of 7 μm diameter and graphite feeder electrodes were purchased from Goodfellow Cambridge. The feeder electrodes with and without reference electrode (Ag/AgCl) were connected to a PalmSens potentiostat/galvanostat for electrochemical measurements. Actuation was observed by a video microscope composed of an IDS camera, a C-mount objective, and a macro extension tube. Micrographic movies and results of the electrochemical measurements were simultaneously recorded by ScreenRec software. The chemical features of the carbon fiber surface were studied by XPS measurements (PHI5000 VersaProbe spectrometer, Chanhassen, MN, USA), with monochromatic Al Kα radiation (hν = 1486.6 eV). The surface morphology was characterized by SEM (FEI Nova NanoSEM 450 outfitted with GENESIS software and an EDX detector). AFM topography images were recorded using a MultiMode AFM with a Nanoscope V controller (Bruker) in tapping mode and analyzed with NanoScope Analysis software.

## Data availability

Source data are provided with this paper. Numerical data generated in this study are provided in the Source Data file. The authors declare that the main data supporting the findings of this study are available within the article and its Supplementary Information files. Additional data, including raw data files from apparatus data acquisition software, are available from the corresponding authors upon request. Source data are provided with this paper.

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

## Acknowledgements

This research was funded in whole by the National Science Center (NCN), Poland, through grant 2022/46/E/ST4/00457 (W.N.). We are grateful to Dr. Piyush Sindhu Sharma for providing smooth carbon fiber. Vishal Shrivastav acknowledges the funding from the European Union's Horizon 2020 research and innovation program under the Marie Skłodowska-Curie grant agreement No 847639 and from the Ministry of Education and Science.

## Author contributions

B.G. conceptualized the project, assembled the experimental setup, conducted the actuation experiments, and wrote different drafts of the manuscript, V.S. contributed to electrochemical actuation experiments and analysis, S.S. contributed to data analysis, discussion, and manuscript reviewing, A.A.M. contributed to actuation experiments and discussion, M.H. performed the XPS measurement and analysis, A.K. contributed in discussion, and data analysis, W.N. conceptualized the project, carried out the SEM and AFM analysis, discussion, manuscript reviewing and editing. All authors discussed the data and commented on the manuscript.

## Competing interests

The authors declare no competing interests.
