## [Transparent Peer Review file · Nature Communications]

Bipolar electrochemical tweezers using pristine carbon fibers with intrinsically asymmetric features

Corresponding Author: Dr Wojciech Nogala

Version 0:

Reviewer comments:

Reviewer #1

(Remarks to the Author)

The manuscript "Bipolar electrochemical tweezers using pristine carbon fibers with intrinsically asymmetric features" shows an interesting concept get bending displacements in wireless departments using LiClO₄ aqueous electrolytes with benzoquinone/hydroquinone as redox couples. Such concept indeed is novel. There are some parts needs more clarity as well other parts needs more characterizations of those carbon fibers with rough morphologies after actuation.

1. The redox-reaction of benzoquinone/hydroquinone triggers the change of carbon fiber surfaces being more rough as main source for the authors actuation mechanism that ClO₄⁻ ions incorporated and the fiber bends. Its well known for carbon fibers in view of EDL process the pores of such carbon fiber plays a vital role. Does such additional oxidation on the surface not induce more pores and are those pores able to detect over BET measurements? If yes please include those.

2. Its also know that sometimes also shown by non wireless operation of MWCNT fiber in electrolyte solution some ions are incorporate and stayed during discharging, triggering cation addition forming the EDL and that fiber moves in opposite direction showing mixed actuation (Shown for TBAPF₆ in ACN). Hence its not exactly the same but if there are grooves and pores in the carbon fiber surface certainly some ions might stuck in and the uniform displacement might be interrupted. Did the authors considering such?

3. If the surface of carbon fibers changed with those additional oxidation also those becomes more rough, would Raman not be able to identify such changes and therefore verify that those additional grooves exist? The SEM image showing such are far less convincing at least a higher resolution maybe TEM images should be provided. Please add those

4. The other part of possible application as an electrolyte cell is needed for such actuation are rather limited as well the bending is quite low. Please also include how much is the bending in mm or micrometer, what is the speed of such and long term measurements if such are made? Did the authors made more than one sample to test or several and are those results reproducible? Please comment such.

5. What is the LiClO₄ concentration and would higher or lower change the outcome? The parameter from where such beding depends needs to be included. Does the surface conductivity of carbon fiber plays a role and if those defects implemented are the bending of the carbon fiber influenced? Please comment on such

Minor parts: Page 3 line 81 What means (590)?

Reviewer #2

(Remarks to the Author)

Gupta et al. present a work of electrochemically driven actuators using bipolar electrochemistry, that can manipulate carbon fibers as a tweezer at certain potentials. The topic as well as the results are quite interesting, but this reviewer has some concerns about the conclusions extracted from limited experimental data. Therefore, major revisions are required prior to publication.

According to the authors when more than 2V is applied to the fiber with non-uniform grooves, a displacement of the fiber resembling to arm movement is observed. Similar displacement is not observed on the fiber with uniform grooves. However,

authors do not make an effort to provide the difference between uniform vs non-uniform grooves on the fibers, since cross-section of carbon fibers on provided SEM images are almost the same. Authors should be able to provide higher magnification images and measure the roughness of the fibers to prove their point (perhaps AFM will do a better job). The schematic illustration on Fig 2 is not supported experimentally. If there is a circular geometry difference (x/y) that is responsible for the actuation, making fibers with more than 10% x/y ratio would be more informative for the scientific community. The fact the actuation occurs at higher voltages suggest that water oxidation has major contribution, and further experimentations can really help to elucidate. Furthermore, authors suggest that C oxidation occurs in the process and XPS is employed to evaluate its degree of oxidation. Authors should plot similar ranges of binding energies in order to facilitate its observation.

Other minor things that need to be addressed:

Some employed acronyms are not defined in text at all, such as the HQ and BQ redox couple but they are employed in figures

Videos in supporting information (SI) are not properly labeled. While SI refers as Video S1, S2, S3, . . . , the actual videos have different names

Authors need revise their references. Some titles do not correspond to the references listed by authors. Such as Ref 23, the Ref can be found in literature however it has a different title to the one listed on the manuscript.

Reviewer #3

(Remarks to the Author)

This paper is based on the principle of closed bipolar electrodes and the observation of the actuator behaviour of carbon fibres. The device is of great interest because it has the potential to be driven as wireless tweezers without direct power supply. An explanation of the necessity of using closed bipolar electrochemistry in this study, including the authors' previous work, would be appreciated, as how to use open or closed bipolar electrochemistry, especially in terms of solution resistance, should be very different.

As the mechanism of the actuation described in Figure 3 and related text is complex, it would be easier to understand if more detailed model diagrams or other information were available. The critical point in the series of mechanisms is the oxidative modification of the carbon fibre surface, which also means that the system is irreversible.

Indeed, even in the narrower potential window, the gradual decrease of current was observed in Figure 6a.

As long as this principle is used, it will be difficult to see it as an actuator.

In Figure 1, the authors claim the asymmetry of distribution of the grooves for each carbon fiber, but I don't find the asymmetry in the images.

Version 1:

Reviewer comments:

Reviewer #1

(Remarks to the Author)

The authors made revision, answer all question and add new explanation. The manuscript can be accepted

Reviewer #2

(Remarks to the Author)

This reviewer remains unconvinced about the proposed activation mechanism. The provided AFM images are of insufficient quality to support authors's assesment. In fact the color based scales for both images are extremely different: (g) with a range of 118 nm while (h) with 2396 nm range, if any this would not support their claim.

Reviewer thinks that attributed asymmetry of the fiber cross-section responsible for actuation is extremely low, not evident in the SEM images. Have the authors tried polishing on side of the fiber in order to modify the geometry of the cross-section and increase the asymmetry?

Reviewer #3

(Remarks to the Author)

Considering the revised version and answers to the queris by reviewers, this manuscript has been well improved and potentially acceptable for publication.

Version 2:

Reviewer comments:

Reviewer #2

(Remarks to the Author)

Considering the revised version and the author's responses to the reviewers' comments, the manscript has been

significantly improved and now is potentially suitable for publication

We are grateful for all the *Reviewer's comments, suggestions, and indications of errors*. Below we addressed these comments in **bold**. Modified parts of the revised manuscript are highlighted in **yellow**.

Reviewer #1 (Remarks to the Author):

The manuscript "Bipolar electrochemical tweezers using pristine carbon fibers with intrinsically asymmetric features" shows an interesting concept that get bending displacements in wireless departments using LiClO₄ aqueous electrolytes with benzoquinone/hydroquinone as redox couples. Such a concept indeed is novel. There are some parts that need more clarity as well other parts need more characterizations of those carbon fibers with rough morphologies after actuation.

1. The redox-reaction of benzoquinone/hydroquinone triggers the change of carbon fiber surfaces being rougher as main source for the authors actuation mechanism that ClO₄⁻ ions incorporated and the fiber bends. It's well known for carbon fibers in view of EDL process and the pores of such carbon fiber play a vital role. Does such additional oxidation on the surface not induce more pores and are those pores able to be detected over BET measurements? If yes, please include those.

It is true that the porosity of carbon fiber increases upon oxidation. One can verify this phenomenon by measuring specific surface area, e.g., with BET. However, for BET characterization, if one needs to use at least ~0.1 g of material. For a density of 2 g/cm³, it would be necessary to modify and analyze ca. 640 meters of 10 μm diameter fiber. Due to experimental difficulties related to the modification and analysis of such long fiber, we performed alternative analyses, i.e., EDL capacitance measurements by electrochemical impedance spectroscopy (EIS) at different DC potentials as well as ex-situ analysis of the fiber surface topography with atomic force microscopy (AFM). Indeed, EDL capacitance and surface roughness increase after anodic polarization. (Figure 3). This confirms that actuation coincides with the increase in porosity in the carbon fiber. Below is the modified part of the discussion in the revised manuscript and new Figure 3.

With the increase in potential, it is assumed that the porosity increases, which is responsible for the larger value of double-layer capacitance, as shown in **Figure 3 c. This feature is further confirmed by electrochemical impedance spectroscopy measurements at different**

potentials. It was found that an increase in double-layer capacitance with an increase in electric potential is due to an increase in porosity.

Figure 3: AFM image of asymmetric carbon fiber a) before and b) after bipolar electrochemical oxidation from 0-3 V in 10 mM BQ/HQ and 0.25 M LiClO₄. c) EDL capacitances derived from impedance spectra recorded with a bundle of fibers at various DC potentials.

This feature is further confirmed by AFM measurement before and after the electrochemical actuation of carbon fiber, as shown in **Figures 3 a and b**. The surface roughness of carbon fiber increases after electrochemical oxidation. Nanoscale roughness factors before and after oxidation are 1.01 and 1.06 nm, respectively.

2. Its also know that sometimes also shown by non-wireless operation of MWCNT fiber in electrolyte solution some ions are incorporate and stayed during discharging, triggering cation addition forming the EDL and that fiber moves in opposite direction showing mixed actuation (Shown for TBAPF6 in ACN). Hence its not exactly the same but if there are grooves and pores in the carbon fiber surface certainly some ions might stuck in and the uniform displacement might be interrupted. Did the authors consider such?

Indeed, in our observations, the actuation happens in the positive direction of the potential scan. During the reverse cycle, it decreases initially and then increases. The mechanism suggested by the Reviewer observed before in the case of MWCNT fits the behavior of carbon fiber. Possibly, stuck ClO₄⁻ ions are not expelled in the reverse potential sweep, but their charge becomes counterbalanced by the insertion of Li⁺ cations. The revised manuscript is updated accordingly by discussion and reference.

Another intriguing feature can be seen during the reverse potential sweep. In the case of reversible charging-discharging of the EDL, only ClO₄⁻ ions would desorb from the fiber, causing recovery of its initial position during the voltage scan from 3 V to 2 V. However,

reactuation is observed when the voltage is further decreased from 2 to 0 V. Position of the fiber is not fully recovered probably because of thermodynamically irreversible oxidation of carbon fiber causing the hysteresis. Li^+ ions can enter the EDL with entrapped ClO_4^- ions during the reverse sweep (from 2 V to 0 V), resulting in an increase of deflection higher than the one originating from the initial insertion of ClO_4^- . Similar behavior was observed in the case of multiwall carbon nanotube fiber, where anions incorporated into the fiber stayed during discharging, triggering cation addition to the EDL, and the fiber showed mixed actuation²⁶. The concentration of LiClO_4 does not affect actuation and current (**Supporting information Figure S2**). The magnitude of actuation depends on the applied potential and the length of the fiber. Strong actuation also occurs at voltages lower than 0 V (**supporting information Video S3**), which further confirms the role of Li^+ ions uptake.

c)

d)

Figure 4: a) A scheme of the three-electrode set-up in a closed bipolar electrochemical cell used for carbon fiber voltammetry studied during the actuation b) Voltammograms of carbon fiber during its upward/downward deflection with ionic uptake in a specific potential region c) deflection at the different voltage transitions (2-3 V and 2-0 V), d) corresponding to the ionic transfer process in and out of EDL.

3. If the surface of carbon fibers changed with those additional oxidations also those becomes rougher, would Raman not be able to identify such changes and therefore verify that those additional grooves exist? The SEM image showing such are far less convincing, at least a higher resolution maybe TEM images should be provided. Please add those

It is true that the contrast in SEM images is not only from the topography of the surface. Other properties affect the intensity of the secondary electrons emitted by the studied sample. Raman spectroscopy/microscopy could identify the chemical structure of the carbon fiber rather than additional grooves. Transmission electron microscopy is also not appropriate to study surface features at micrometer size fiber. We carried out AFM analysis to investigate microscopic changes in the fiber after electrochemical oxidation and actuation. Obtained AFM images before and after electrochemical oxidation, added to the revised manuscript as Figure 3 a,b, clearly revealed that the surface roughness increases after electrochemical oxidation. The associated discussion is added to the revised manuscript (see reply to comment no. 1.)

4.The other part of possible application as an electrolyte cell is needed for such actuation are rather limited as well the bending is quite low. Please also include how much is the bending in mm or micrometer, what is the speed of such and long term measurements if such are made? Did the authors made more than one sample to test or several and are those results reproducible? Please comment such.

We agree with the Reviewer that an electrolyte cell required for actuation limits possible applications to an electrolyte environment only. The revised supplementary information file is updated with a scaled optical micrograph (Figure S1) of fiber before actuation with an overlaid image after actuation. The maximum actuation achieved with carbon fiber was 1.2 mm at 3 V potential. The speed of achieving deflection was synchronized with the potential scan rate applied. We carried out studies at a scan rate of 50 mV/s. Actuation of 0.55 mm occurring between ~2.2 V to 3 V took ~16 seconds. This corresponds to an average speed of ~34 $\mu\text{m/s}$. The fastest average speed we observed

when pulsing the potential between 0 V and 3 V (Video S4) was around 1.1 mm/s. We made multiple experiments with the actuation of different fibers. The actuation always occurs, however, its direction depends on the angular alignment of the fiber (see Video S2 and S3). The revised manuscript is updated with the most important information regarding this point.

The total deflection in carbon fiber is 0.55 mm (Supporting information Figure S1) from 2 V to 3 V. Maximum deflection occurs at 3 V. Above 3 V, oxygen bubble generation takes place, which creates ambiguous deflection.

Figure S1. Image of actuation in carbon fiber. 1) polarized at 2 V and 2) at 3 V.

5. What is the LiClO_4 concentration and would higher or lower change the outcome? The parameter from where such bending depends needs to be included. Does the surface conductivity of carbon fiber plays a role and if those defects implemented are the bending of the carbon fiber influenced? Please comment on such

The concentration of LiClO_4 was 0.25 M. It was mentioned only in the experimental section of the original submission. The concentration is added in Figure 3 caption in the revised manuscript. As per the suggestion of the Reviewer, we performed additional control experiments with LiClO_4 concentrations of 0.1 M, 0.25 M and 0.5 M. However, the recorded currents (Figure S2 in Supporting Information), as well as the deflections of the fiber observed through a stereo microscope, were found to be the same for all three tested concentrations. This observation suggests that the mass transport (migration and diffusion) of both ClO_4^- and Li^+ ions is not a factor limiting the rate of the

studied process. This also suggests that the amount of ions inserted into the grooves upon polarization depends on the applied potential, not on the concentration of ions in the electrolyte. We did not perform control experiments with a concentration lower than 0.1 M to avoid an ohmic potential drop, which could be caused by increased resistivity of diluted electrolyte. In general, the bending magnitude of a certain fiber depends on the applied potential and the length of the fiber. The revised manuscript is updated with additional explanation (cited below). We do not expect any influence of surface conductivity, as the carbon fiber material is electronically conductive in its bulk. The core of the fiber acts as a current collector. Even for resistive surfaces, the electrons for charging the EDL are delivered and received to and from the surface by the fiber core.

The concentration of LiClO_4 does not affect actuation and current (**Supporting information Figure S2**). The magnitude of actuation depends on the applied potential and the length of the fiber.

Figure S2. Cyclic voltammograms of carbon fiber at different concentrations of LiClO_4 .

Minor parts: Page 3 line 81 What means (590)?

In the original submission, 590 was a revolution rate from ref. 21. As this number is not crucial in this work, it is not mentioned in the revised manuscript.

...reversible multiple revolutions²¹.

Reviewer #2 (Remarks to the Author):

Gupta et al. present a work of electrochemically driven actuators using bipolar electrochemistry, that can manipulate carbon fibers as a tweezer at certain potentials. The topic as well as the results are quite interesting, but this reviewer has some concerns about the conclusions extracted from limited experimental data. Therefore, major revisions are required prior to publication.

According to the authors when more than 2V is applied to the fiber with non-uniform grooves, a displacement of the fiber resembling to arm movement is observed. Similar displacement is not observed on the fiber with uniform grooves. However, authors do not make an effort to provide the difference between uniform vs non-uniform grooves on the fibers, since cross-section of carbon fibers on provided SEM images are almost the same. Authors should be able to provide higher magnification images and measure the roughness of the fibers to prove their point (perhaps AFM will do a better job).

We agree with the Reviewer. The original submission lacked detailed information on the surface structure of the fibers. The revised manuscript is updated with AFM images (Fig. 1. g,h) of grooved and smooth fibers. The grooved fiber surface possesses a clearly visible nonuniform surface in comparison to the surface of the smooth fiber. Besides AFM images, the revised manuscript is updated with appropriate sentences.

Additionally, AFM was carried out to investigate the microscopic features of the grooves. The rough fiber possesses clearly visible nonuniform surface features (Fig. 1 g) in comparison to the rather uniform surface of a smooth fiber (Fig. 1 h).

Figure 1: Scanning electron microscopy images of a carbon fiber surface for 0° and 180° axial rotation of a) smooth carbon fiber and b) rough carbon fiber; cross-section view of c) a smooth carbon fiber and f) a rough carbon fiber, g) AFM image of rough carbon fiber, and h) AFM of smooth carbon fiber.

The schematic illustration on Fig 2 is not supported experimentally. If there is a circular geometry difference (x/y) that is responsible for the actuation, making fibers with more than 10% x/y ratio would be more informative for the scientific community.

We agree with the Reviewer that fibers with higher asymmetry would be more informative. In this study, we used commercially available fibers as received and showed the ability of the grooved fiber to actuate upon polarization. The asymmetry of the grooved fiber results from the manufacturing process. Perhaps a worn nozzle used in the extruder used during manufacturing was a source of asymmetry. A smooth fiber with a more circular cross-section does not exhibit actuation. We made an attempt to enhance the asymmetry by squeezing the fiber between two glass slides. However, the fiber does not possess appropriate plasticity. This attempt resulted in cracked fiber. A lack of plasticity is also observed when bending the fiber mechanically. It always regains its initial shape after releasing the strain or brakes if the bending strain is too high. The actuation observed with fiber whose non-uniform geometry is confirmed by microscopic techniques (AFM of the lateral curved surface and SEM of cross-section) in comparison to a lack of actuation of a uniform control fiber confirms a crucial role of the asymmetry in the actuation. More asymmetric fibers could be studied after their intentional manufacturing, e.g., with an asymmetric extruder nozzle. In our opinion, studies on the actuation of such manufactured carbon fibers are beyond the discovery presented in this communication. We suggested this in the last sentence of the conclusions in the revised manuscript.

As a result, these findings may open up intriguing possibilities for **actuators based on pre-fabricated asymmetric carbon fibers.**

The fact the actuation occurs at higher voltages suggest that water oxidation has major contribution, and further experimentations can really help to elucidate. Furthermore, authors suggest that C oxidation occurs in the process and XPS is employed to evaluate its degree of oxidation. Authors should plot similar ranges of binding energies in order to facilitate its observation.

It is true that water oxidation is driven at potentials where actuation occurs. For potentials higher than 3 V oxygen evolution is fast enough to generate bubbles. Then the fiber moves upward due to buoyancy caused by attached gas bubbles. However, before gas evolution starts, depending on the angular alignment of the fiber, it can move downwards (Video S2). This confirms that the actuation is not due to water oxidation but due to oxidation of the fiber surface and uptake of ClO_4^- anions. Figure 5 in the revised

manuscript is updated. The XPS spectra are plotted for similar ranges of binding energies.

Figure 5: a) Survey b) C1s c) O1s XPS spectra of a pristine and treated carbon fiber followed by deconvolution and d) Cl 2p spectra of treated carbon fiber.

Other minor things that need to be addressed:

Some employed acronyms are not defined in text at all, such as the HQ and BQ redox couple but they are employed in figures

These acronyms are defined in the revised manuscript.

... benzoquinone (BQ) and hydroquinone (HQ) as a redox couple, ...

Videos in supporting information (SI) are not properly labeled. While SI refers as Video S1, S2, S3, . . . , the actual videos have different names

The video descriptions are corrected in the Supporting Information file.

Authors need revise their references. Some titles do not correspond to the references listed by authors. Such as Ref 23, the Ref can be found in literature however it has a different tittle to the one listed on the manuscript

References are corrected.

Reviewer #3

This paper is based on the principle of closed bipolar electrodes and the observation of the actuator behaviour of carbon fibres. The device is of great interest because it has the potential to be driven as wireless tweezers without direct power supply. An explanation of the necessity of using closed bipolar electrochemistry in this study, including the authors' previous work, would be appreciated, as how to use open or closed bipolar electrochemistry, especially in terms of solution resistance, should be very different.

A closed bipolar cell is necessary for wireless actuation with low power consumption. One can perform actuation in an open bipolar cell (ref. 10-12). Then the actuating object is polarized by an electric field (potential gradient) in the electrolyte. The majority of current between feeder electrodes passes through the electrolyte and the current passing through the actuating object comprises only a small fraction of the total current. Then, one needs to use a power supply able to deliver enough voltage and current. For an actuation in a closed bipolar cell, a standard low-power potentiostat is enough. Then, the entire current flowing between the feeder electrodes flows through the actuating object, and the process is highly energy-efficient.

Question 1 As the mechanism of the actuation described in Figure 3 and related text is complex, it would be easier to understand if more detailed model diagrams or other information were available. The critical point in the series of mechanisms is the oxidative modification of the carbon fibre surface, which also means that the system is irreversible.

We agree with the Reviewer that the actuation mechanism follows oxidative modification of the carbon fiber, which causes the irreversibility of the actuation. The mechanism is simplified based on the ClO_4^- ion uptake during the oxidation cycle (forward cycle) and Li^+ ion uptake during the reduction cycle (reverse cycle). In the revised manuscript, the Figure has been modified accordingly. Its updated number is 4.

Figure 4: a) A scheme of the three-electrode set-up in a closed bipolar electrochemical cell used for carbon fiber voltammetry studied during the actuation b) Voltammograms of carbon fiber during its upward/downward deflection with ionic uptake in a specific potential region c) deflection at the different voltage transitions (2-3 V and 2-0 V), d) corresponding to the ionic transfer process in and out of EDL.

Question: Indeed, even in the narrower potential window, the gradual decrease of current was observed in Figure 6a.

In the revised manuscript, this figure has number 7. The gradual decrease of the cathodic (negative) current at the initial stages of reverse pulses (to 0 V) is due to the fact that the fiber was held actuated for a long time at +3 V before starting the sequence of pulses. Holding at +3 V caused exhaustive oxidation of the fiber surface and the

formation of pores (see revised Figure 3). Its complete reduction could not occur during a few initial pulses to 0V. One can see that this concerns only the initial stages of the reverse pulses. At the end of each step, the current is similar for all pulses. For reversible amperometric behavior, the system needs more pulses for stabilization.

As long as this principle is used, it will be difficult to see it as an actuator. In Figure 1, the authors claim the asymmetry of distribution of the grooves for each carbon fiber, but I don't find the asymmetry in the images.

The asymmetry in terms of grooves is not high when using simple geometric parameters. However, a small asymmetry is enough to cause actuation. In Fig. 1 c), one can see that the lower half of the fiber cross-section possesses a noticeably different geometry of grooves than the upper half. Thus, the lateral microscopic surface area of one side of the fiber differs from the lateral microscopic area of the other side of the fiber. AFM images in the revised Figure 1 (pasted in the answer to the Reviewer #2 comment) also show the lateral surface of the fiber is non-uniform in comparison to the lateral surface of a smooth fiber. The discussion in the revised manuscript is updated for better understanding.

The cross-section image (Fig. 1 c) clearly illustrates an asymmetric distribution of these grooves. One can observe that the lower half of the fiber has a noticeably different groove geometry than the upper half. Consequently, the lateral microscopic surface area on one side of the fiber differs from that on the other side. ... Additionally, AFM was carried out to investigate the microscopic features of the grooves. The rough fiber possesses clearly visible nonuniform surface features (Fig. 1 g) in comparison to the rather uniform surface of a smooth fiber (Fig. 1 h).

We are grateful for the Reviewer's #1 and #3 for finding our previous manuscript improved and for the Reviewer's #2 comments and suggestions. Below we addressed these comments in **bold**. Modified parts of the revised manuscript are highlighted in **yellow**.

Reviewer #2 (Remarks to the Author):

This reviewer remains unconvinced about the proposed activation mechanism. The provided AFM images are of insufficient quality to support authors's assesment. In fact the color based scales for both images are extremely different: (g) with a range of 118 nm while (h) with 2396 nm range, if any this would not support their claim.

We agree with the Reviewer that the AFM images (Fig. 1(i) and (j) in the revised manuscript) of the convex side surfaces of the carbon fibers do not exhibit extreme differences in surface topography. However, we maintain that, in contrast to the AFM image of the smooth fiber, grooves are visible on the rough fiber. For convenience, the images are provided below.

Our intention was to present unprocessed (as-recorded) images without any additional post-processing. The actual range of the false color scale in Fig. 1(i) is not 118 nm. It is $359 - (-477) = 836$ nm. The Z-scale range of 2396 nm in Fig. 1(j) in the revised manuscript is greater than the 836 nm range in the image of the rough fiber due to the smaller radius of curvature of the smooth fiber. The rough fiber has a diameter of $10\ \mu\text{m}$, whereas the smooth fiber has a diameter of $7\ \mu\text{m}$. Both AFM images depict the same projected surface area of $(5 \times 5)\ \mu\text{m}^2$. It is expected that the Z-scale range of the image showing a fiber with a smaller diameter would span a larger range. The Experimental section of the revised manuscript has been updated to include the previously missing nominal diameter of the smooth fiber.

(The rough carbon fibers of $10\ \mu\text{m}$ diameter and smooth of $7\ \mu\text{m}$ diameter and graphite feeder electrodes were purchased from Goodfellow Cambridge).

The revised supporting information file is supplemented with cross sections of AFM images from Figure 1.

Figure S1. Cross sections extracted from AFM images in Fig. 1 i) (upper plot) and j) (lower plot). Positions of cross-section lines are shown in insets.

Despite the larger Z-range for the smooth fiber caused by its lower diameter, one can see that the rough fiber possesses grooves. The measured depths of the selected topographic features also differ substantially.

Reviewer thinks that attributed asymmetry of the fiber cross-section responsible for actuation is extremely low, not evident in the SEM images. Have the authors tried polishing

on side of the fiber in order to modify the geometry of the cross-section and increase the asymmetry?

We are grateful for this suggestion. We made several unsuccessful attempts to polish one side of the fiber. Although the idea is interesting, we were unable to polish the sides of carbon fibers over a length of at least 1 cm, which is necessary for subsequent actuation experiments. The fibers are too fragile to withstand mechanical polishing. We also attempted to compress an individual fiber between two glass slides. Similar to the bending test, the fibers showed no plasticity and disintegrated into particles upon compression. Additionally, we placed fibers lying on a glass slide inside an oxygen plasma chamber. This resulted in uniform etching and a slight reduction in diameter, but without any added asymmetry in the cross-section. The only successful method we tested for increasing asymmetry was sputtering a gold layer onto one side of the fibers. Screenshots of a preliminary actuation experiment and an SEM image of the Au-modified fiber are provided below.

In this case, the intentionally introduced asymmetry is chemical rather than geometric, and thus falls outside the scope of this article. The revised manuscript does not include data on Au-modified fibers; we share this information solely in response to the Reviewer's comment.

In light of the Reviewer's concern regarding the lack of visible asymmetry in the SEM images, we recorded additional SEM images of fiber cross sections. These were not obtained by polishing a composite of carbon fibers embedded in a glass capillary, but by aligning the fibers parallel to the electron beam within the electron microscope chamber. This method enabled the acquisition of clearer SEM images. In previous attempts, the surrounding non-conductive glass and polishing debris pressed into the carbon material hindered the contrast necessary to distinguish features within the fiber cross sections.

SEM images of mechanically cut fibers, properly aligned in the electron microscope chamber, reveal an asymmetric distribution of pores within the rough fibers with grooves, while the smooth fibers exhibit a more uniform internal structure. Such internal features cannot be observed with AFM, as the AFM tip is not narrow enough to penetrate the pores. Figure 1 in the revised manuscript has been updated with selected images (g, h), and the text has been modified accordingly.

The cross-section image of the fiber fused in a glass capillary, followed by polishing (Fig. 1c), clearly illustrates an asymmetric distribution of these grooves. One can observe that the lower half of the fiber has a noticeably different groove geometry than the upper half. Consequently, the lateral microscopic surface area on one side of the fiber differs from that on the other side. Moreover, the cross section of fiber exposed not by polishing of a composite with a glass capillary, but by mechanical cutting (Fig. 1g), shows a clear asymmetric pore distribution, which is substantially different from the cross section of a smooth fiber. The latter one is compact and symmetrical (Fig. 1h).

Figure 1: Scanning electron microscopy images of a carbon fiber surface for 0° and 180° axial rotation of rough carbon fiber a),b) and smooth carbon fiber d),e); cross-section view of carbon fiber fused in glass capillary and exposed by polishing: c) a rough carbon fiber, and f) a smooth carbon fiber. Figures g) and h) show cross-sections of rough and smooth carbon fibers, respectively. i) AFM image of rough carbon fiber, and j) AFM image of smooth carbon fiber. Cross sections of AFM images are presented in the supporting information, Figure S1.

The SEM images presented in Fig. 1(g) and (h) are not exceptional. We recorded many similar images of mechanically cut and properly aligned fibers.

Selected SEM images of cross-sections of rough fibers:

Selected SEM images of cross-sections of smooth fibers: